# Preserving a robust CsPbI$_3$ perovskite phase via pressure-directed octahedral tilt

Feng Ke[1,2,5], Chenxu Wang [2,5], Chunjing Jia [1,5], Nathan R. Wolf [3], Jiejuan Yan[2], Shanyuan Niu [1,2], Thomas P. Devereaux [1,4], Hemamala I. Karunadasa[1,3], Wendy L. Mao [1,2] & Yu Lin [1✉]

Functional CsPbI$_3$ perovskite phases are not stable at ambient conditions and spontaneously convert to a non-perovskite δ phase, limiting their applications as solar cell materials. We demonstrate the preservation of a black CsPbI$_3$ perovskite structure to room temperature by subjecting the δ phase to pressures of 0.1 – 0.6 GPa followed by heating and rapid cooling. Synchrotron X-ray diffraction and Raman spectroscopy indicate that this perovskite phase is consistent with orthorhombic γ-CsPbI$_3$. Once formed, γ-CsPbI$_3$ could be then retained after releasing pressure to ambient conditions and shows substantial stability at 35% relative humidity. First-principles density functional theory calculations indicate that compression directs the out-of-phase and in-phase tilt between the [PbI$_6$]$^{4-}$ octahedra which in turn tune the energy difference between δ- and γ-CsPbI$_3$, leading to the preservation of γ-CsPbI$_3$. Here, we present a high-pressure strategy for manipulating the (meta)stability of halide perovskites for the synthesis of desirable phases with enhanced materials functionality.

[1] Stanford Institute for Materials and Energy Sciences, SLAC National Accelerator Laboratory, Menlo Park, CA 94025, USA. [2] Department of Geological Sciences, Stanford University, Stanford, CA 94305, USA. [3] Department of Chemistry, Stanford University, Stanford, CA 94305, USA. [4] Department of Materials Science and Engineering, Stanford University, Stanford, CA 94305, USA. [5] These authors contributed equally: Feng Ke, Chenxu Wang, Chunjing Jia. ✉email: lyforest@stanford.edu

Lead halide perovskite solar cells based on (MA)PbI₃ (MA = CH₃NH₃⁺) have achieved power-conversion efficiencies comparable to commercial Si-based solar cells in recent years[1–3]. However, the volatile organic MA cation is largely responsible for this material's instability to humidity and heat, a critical issue that hinders its large-scale implementation. Substitution of an inorganic cation such as Cs⁺ has been explored as a way to improve the robustness of lead halide perovskites[4–7]. There are four known polymorphs of CsPbI₃[8–13]: a room-temperature non-perovskite phase (δ), and three high-temperature perovskite-related phases with cubic (α), tetragonal (β), and orthorhombic (γ) structures. Perovskite-structured CsPbI₃ has a bandgap of 1.6–1.8 eV that is favorable for photovoltaic applications[4–9]. However, a major challenge to realizing CsPbI₃-based solar cells is that these black perovskite phases are not thermodynamically stable at ambient conditions and spontaneously convert to the non-functional δ phase[10–13]. A recent study reported that perovskite solar cells made from surface-treated β-CsPbI₃ achieved power-conversion efficiencies >18%, but only at high temperature[4]. It is crucial to find ways to stabilize CsPbI₃ perovskites at room temperature for practical device operation.

Previous studies have offered a few strategies for stabilizing CsPbI₃ perovskite phases to room temperature, including thermal engineering[9–13], compositional tuning[14–17], nanocrystal growth[5,18,19], solvent and surface treatments[6–8,20–28], and strain engineering[29]. However, these approaches present several drawbacks and limitations[30]. For instance, thermal engineering based on a solid-state method required rigorously anhydrous reagents, a moisture-free environment, a melt state at high temperature, and extremely rapid cooling. The as-preserved bulk γ-CsPbI₃ also showed severe moisture sensitivity[13]. The strain-stabilized γ-CsPbI₃ film was rendered by the use of a combined substrate clamping and biaxial strain in an inert atmosphere and returned back to the δ phase within minutes in air at 27% relative humidity (RH)[29]. Compositional tuning introduced undesirable changes in the electronic structure[20,21], and nanomaterials showed an increase in grain boundaries which inhibited charge transport and caused considerable recombination loss[6,15]. These drawbacks motivate us to explore other approaches.

Previous calculations suggested that the [PbI₆]⁴⁻ inter-octahedral tilt had a large influence on the formation energies of the three perovskite phases (α, β, and γ) of CsPbI₃, among which the γ phase has the lowest energy owing to its largest octahedral tilt[9]. Hence, tuning the octahedral tilt of the high-temperature perovskite phase(s) using external stimuli such as pressure will be favorable for preserving the desired structure back to ambient conditions. Many halide perovskite systems have been studied at high pressure. Pressure has proven to be an effective and clean tool for modifying their structures and creating exotic physical properties[31–35], such as the bandgap modulation[31,32] and closure in compressed (MA)PbI₃[35,36] and emission enhancement in compressed halide double perovskites[37]. The pressure was also found to markedly tilt the [PbI₆]⁴⁻ octahedra in CsPbI₃ perovskite nanocrystals[38,39]. However, none of the desired high-pressure phases has been preserved back to ambient conditions after releasing pressure.

In this work, we study the solid-to-solid structural evolution of CsPbI₃ at high-pressure and high-temperature conditions in a resistive-heated diamond–anvil cell (Fig. 1a) using synchrotron X-ray diffraction (XRD), Raman spectroscopy, photoluminescence (PL) measurements, and first-principles density functional theory (DFT) calculations. We identify viable pressure–temperature (P–T) pathways to access and quench a black CsPbI₃ perovskite phase back to ambient conditions. The preserved CsPbI₃ phase shows substantial stability to moisture

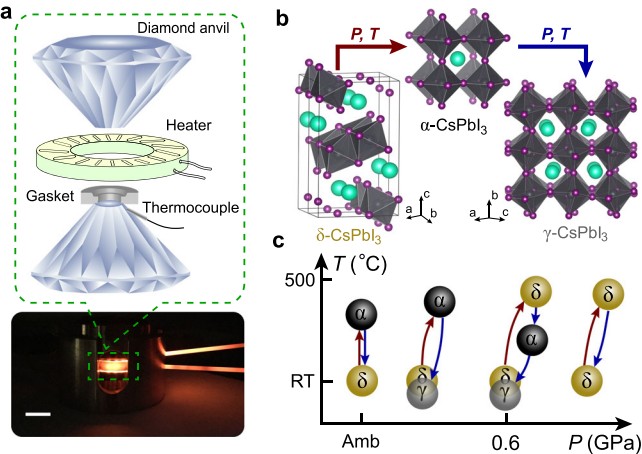

**Fig. 1 Experimental setup and pressure-induced preservation of a metastable γ-CsPbI₃ perovskite to ambient conditions. a** Illustration of a cross-sectional view of the high-pressure and high-temperature setup. The white scale bar is 1 cm. Inset: Schematic of a diamond–anvil cell and an external resistive heater used in our experiments. **b** The observed structures of CsPbI₃ at varying P–T cycles. The red and blue arrows represent the most viable heating and cooling pathways to preserve a γ-CsPbI₃ perovskite phase, where the applied pressure (P) is 0.1–0.6 GPa and temperature (T) is up to 450 °C. Atoms in the structures are shown in gray (Pb), purple (I), and cyan (Cs). **c** Representative P–T phase transformations observed in the study. RT and Amb indicate room temperature and ambient pressure, respectively. The yellow δ and gray γ circles around RT are in fact both at RT and plotted with a slight offset for clarity. Once γ-CsPbI₃ is formed by temperature quenching at high pressure, it can be preserved back to ambient conditions after fully releasing pressure.

and its PL intensity remains above 80% of the initial intensity for more than 30 days and up to 10 days at 20% and 35% RH, respectively.

## Results

**P–T phase diagram of CsPbI₃.** In situ XRD and Raman measurements were conducted to study the structural evolution of CsPbI₃ as a function of pressure along heating and cooling routes. All the diamond–anvil–cell preparation, sample loading, and P–T treatments were done in air without special handling. The samples were first compressed to a target pressure at room temperature followed by heating and cooling from high temperature. The pressure may shift slightly as the temperature is varied during the cycle, but the final pressure after cooling back to room temperature is nearly identical to the initial pressure before heating.

At ambient pressure, XRD indicates that the starting δ-CsPbI₃ phase transforms to the cubic α-CsPbI₃ phase when heated to 320 °C (Fig. 2a), consistent with previous observations[10–12]. With increasing pressure, the δ-to-α phase transition occurs at higher temperatures. Specifically, the transition temperature increases to 370 °C at 0.6 GPa (Fig. 2b) and exceeds 500 °C at 1.1 GPa (Fig. 2c). Raman measurements also agree with the XRD results. At ambient pressure and with heating up to 315 °C, the Raman modes of δ-CsPbI₃ remain nearly invariant except for subtle frequency shifts, peak broadening, and changes in relative peak intensities (Fig. 2d). As the temperature reaches 315 °C, the color of the sample changes dramatically from yellow to black (Supplementary Fig. 1) and all of the Raman modes suddenly disappear, suggesting that CsPbI₃ converts to the cubic α phase which is Raman inactive based on the group theory analysis on a cubic perovskite structure[40,41]. With increasing pressure, the disappearance of the Raman modes (Fig. 2e, f) and the color

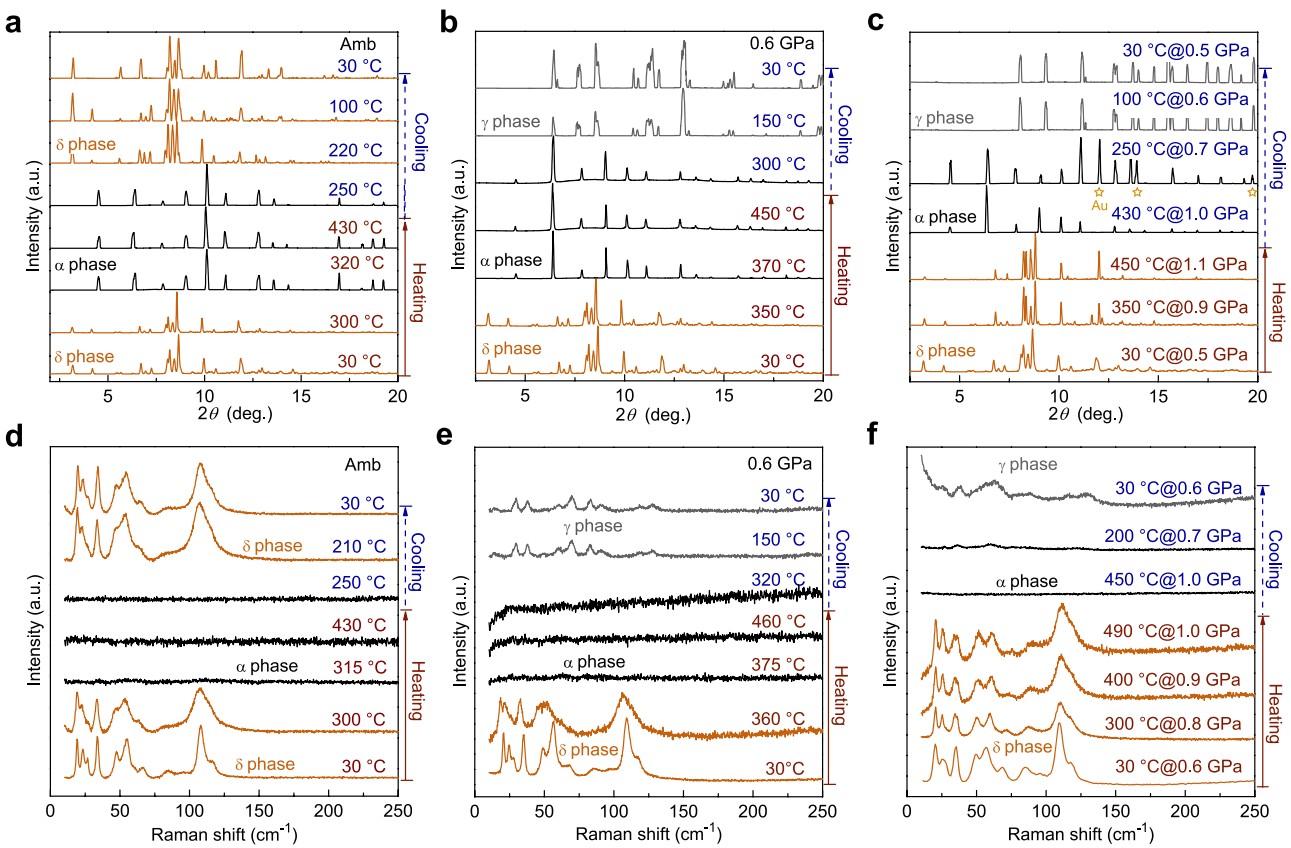

**Fig. 2 X-ray diffraction (XRD) and Raman spectra showing the effect of pressure on the structural transition of CsPbI₃ along with thermal treatments.** **a–c** XRD results ($\lambda = 0.4959$ Å) of CsPbI₃ along heating and cooling cycles at ambient pressure (**a**), 0.6 GPa (**b**), and varying pressures (**c**). The sample remains in a crystalline state up to the highest temperature studied in each cycle. The intensity differences of XRD patterns for α- and γ-CsPbI₃ could be a result of different grain sizes and orientations due to the grain growth to highly textured, coarse grains at high temperatures. **d–f** Raman spectra of CsPbI₃ along heating and cooling cycles at ambient pressure (**d**), 0.6 GPa (**e**), and varying pressures (**f**), confirming the structural transitions observed from XRD measurements.

change from yellow to black occur at higher temperatures, completely consistent with our XRD results. Following slow cooling, the high-temperature α phase transforms back to the δ phase below the transition temperature. Based on the XRD and Raman results, a *P–T* phase diagram of CsPbI₃ was mapped out as shown in Supplementary Fig. 2.

**Pressure effect on preserving γ-CsPbI₃ to ambient conditions**. The structural transitions observed upon rapid cooling at a rate of ~90 K/min strongly depend on the pressure applied to the system. XRD and Raman results indicate that a black perovskite phase can be metastably preserved back to room temperature when the initial δ-CsPbI₃ is subjected to a pressure between 0.1 and 0.6 GPa followed by rapid cooling from high temperature. Three different sets of experimental routes were identified according to the applied pressure: <0.1 GPa (route 1), 0.1–0.6 GPa (routes 2 and 2'), and >0.6 GPa (routes 3, 3', and 3''), as shown in Fig. 3a. At ambient pressure (route 1), when rapidly cooling from 430 °C, the high-temperature α-CsPbI₃ returns to the thermodynamically preferred δ phase below 220 °C (Fig. 2a).

Interestingly, when the applied pressure is between 0.1 and 0.6 GPa (routes 2 and 2'), CsPbI₃ undergoes different structural transitions upon rapid cooling. Taking the cooling process at ~0.6 GPa as an example (Fig. 2b and route 2 in Fig. 3a), as the sample cools from 450 °C, the diffraction peaks of the high-temperature α-CsPbI₃ split obviously below 150 °C, and these changes persist down to room temperature. The preserved diffraction pattern and Raman spectrum differ from those of

the non-perovskite δ phase (Fig. 2b, e). The sample also remains opaque black throughout cooling (Supplementary Fig. 1). In route 2' (Fig. 3a), in which the pressure increases from 0.5 to 1.1 GPa during heating to 450 °C and goes back to 0.5 GPa after cooling to room temperature, XRD and Raman measurements indicate that while CsPbI₃ remains in the yellow δ phase during the entire heating cycle (Fig. 2c, f), the sample instantaneously transforms to the α phase at the onset of cooling, followed by the appearance of a similar black phase as observed in route 2 which persists down to room temperature.

The splitting of diffraction peaks indicates that this preserved black phase has a lower symmetry compared to the cubic α-CsPbI₃ phase. We indexed the diffraction pattern with the other known phases of CsPbI₃ (β, γ, and δ), and found that all of the diffraction peaks of the preserved black phase in Fig. 2b, c could be assigned to the orthorhombic γ-CsPbI₃, but not to β- or δ-CsPbI₃ (Supplementary Figs. 3 and 4). As shown in Supplementary Fig. 5, the diffraction pattern changes from smooth diffraction rings before heating to spots after CsPbI₃ undergoes transitions to the α and γ structures, indicating significant grain growth in the sample to highly textured, coarse grains. The diffraction peak intensities are unreliable as they are mainly determined by the grain size and the orientation of each grain relative to the incident X-ray beam. While all of the peaks in the XRD patterns of the recovered phase can be indexed to γ-CsPbI₃, the relative peak intensities look distinct in routes 2 and 2' (Fig. 2b, c) and are also different from previous studies on powder samples[9,10]. The apparent differences in the diffraction angle (2θ)

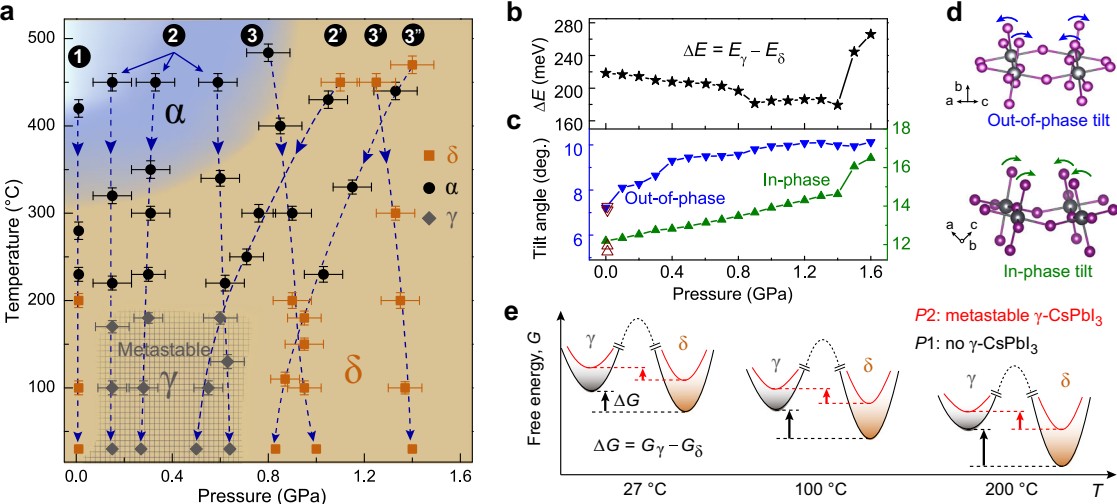

**Fig. 3 Representative cooling routes and the effect of [PbI$_6$]$^{4-}$ octahedral tilt on the energy difference between γ- and δ-CsPbI$_3$ with pressure. a** The experimental cooling routes used to access and preserve γ-CsPbI$_3$ to room temperature. The background color shows the $P$–$T$ phase diagram of CsPbI$_3$ that includes the δ and α phases. γ-CsPbI$_3$ is not a thermodynamically stable phase and only forms during rapid cooling from high temperature marked as a gray cross-hatched region. The numbers inside the black circles indicate different cooling routes. The temperature error bars are determined based on the temperature fluctuation before and after measurements, which further causes the pressure uncertainty (see "Methods" for details). **b** The total energy difference per unit cell between γ- and δ-CsPbI$_3$ ($\Delta E = E_\gamma - E_\delta$) under compression. **c** The evolution of the calculated out-of-phase tilt along [101] and the in-phase tilt along [010] as a function of pressure. The open triangles are the previously reported tilt angles of γ-CsPbI$_3$ at room temperature and ambient pressure[10,13]. **d** Ball-stick models visualizing the out-of-phase and in-phase octahedral tilts. **e** A schematic Gibbs free energy ($G$) diagram of γ- and δ-CsPbI$_3$ as a function of temperature at pressure $P1$ (e.g., 0.8 and 1.6 GPa) without the formation of metastable γ-CsPbI$_3$ (black curve) and at pressure $P2$ (e.g., 1.2 GPa) with the formation of γ-CsPbI$_3$ (red curve). $\Delta G$ ($= G_\gamma - G_\delta$) at 0.8, 1.2, and 1.6 GPa are 0.806, 0.589, and 0.941 eV at 27 °C, 0.934, 0.614, and 1.011 eV at 100 °C, and 1.128, 0.653, and 1.118 eV at 200 °C, respectively.

values of our diffraction peaks from the previous reports[9,10] are due to the different X-ray wavelengths used in the XRD measurements. The Raman spectrum of the preserved black phase also agrees well with that of γ-CsPbI$_3$ synthesized by a solid-state method (Supplementary Fig. 6). No Raman modes or XRD peaks from CsI or PbI$_2$ precursors are observed, indicating no decomposition of CsPbI$_3$.

Our study further indicates that γ-CsPbI$_3$ can only be accessed when the final pressure falls between 0.1 and 0.6 GPa. When the final pressure is above 0.6 GPa, CsPbI$_3$ transforms back to the δ phase after the $P$–$T$ treatments or remains in the δ phase in the entire $P$–$T$ range studied, as supported by three independent routes covering varying $P$–$T$ pathways (routes 3, 3', and 3").

Based on the XRD and Raman results, we determined a $P$–$T$ window marked by a gray cross-hatched area in Fig. 3a for the formation and preservation of γ-CsPbI$_3$. The key ingredient for being able to metastably preserve γ-CsPbI$_3$ to room temperature is the final pressure applied to the system, i.e., 0.1–0.6 GPa. CsPbI$_3$ returns to the non-perovskite δ phase after $P$–$T$ cycles when the applied pressure falls outside this narrow range. The cubic α-CsPbI$_3$ phase is found to serve as an intermediate for accessing γ-CsPbI$_3$. It is encouraging that once preserved to room temperature, γ-CsPbI$_3$ can be retained even after fully releasing pressure back to ambient conditions.

**Relative energetic stability of compressed δ- and γ-CsPbI$_3$.** Although γ-CsPbI$_3$ is a kinetically trapped metastable phase and does not appear on the $P$–$T$ phase diagram, energy calculations can still shed light on the effect of pressure on the relative stability of the competing δ and γ phases (Fig. 3). We first performed first-principles DFT calculations on these two structures as a function of pressure at $T = 0$ K. At 0 GPa, δ-CsPbI$_3$ has a lower total energy compared to γ-CsPbI$_3$, supporting that δ-CsPbI$_3$ is the thermodynamically stable phase. With the application of

pressure, the total energy difference between γ- and δ-CsPbI$_3$ ($\Delta E = E_\gamma - E_\delta$) reduces with pressure (Fig. 3b), indicating an increased stabilization of γ-CsPbI$_3$ with respect to δ-CsPbI$_3$. $\Delta E$ reaches its minimum at 0.9 GPa and remains almost invariant up to 1.4 GPa. With further compression, $\Delta E$ shows a sharp rise due to a larger rate at which the total energy of γ-CsPbI$_3$ increases relative to that of δ-CsPbI$_3$ (Supplementary Fig. 7). DFT results suggest that 0.9–1.4 GPa is the most suitable pressure range for favoring the formation of γ-CsPbI$_3$, spanning a small pressure window of 0.5 GPa as in our experimental results. To further elucidate the combined effect of pressure and temperature on preserving γ-CsPbI$_3$, the Gibbs free energy ($G = E + PV - TS$) that includes the vibrational entropy at varying pressures was calculated (Fig. 3e and Supplementary Fig. 8). It is beyond our computational capacity to survey the entire $P$–$T$ space studied. Representative pressures of 0.8, 1.2, and 1.6 GPa were chosen based on the total energy calculations. The results show that at 1.2 GPa, the Gibbs free energy difference between γ- and δ-CsPbI$_3$ ($\Delta G = G_\gamma - G_\delta$) is much smaller than that at 0.8 and 1.6 GPa, and it keeps reducing as the material cools from 200 °C to room temperature, implying a preferred stabilization of metastable γ-CsPbI$_3$ towards room temperature as pressure falls in the window of 0.9–1.4 GPa.

## Discussion

After analyzing the structural details, the tilt of [PbI$_6$]$^{4-}$ octahedra was found to correlate with the stabilization of the γ phase. There are two types of octahedral tilts in γ-CsPbI$_3$: an out-of-phase tilt along the [101] direction with adjacent octahedra turning in opposite directions and an in-phase tilt along the [010] direction with adjacent octahedra rotating towards the same direction (Fig. 3d and Supplementary Fig. 9). The simulated out-of-phase and in-phase tilt angles at 0 GPa are comparable with the values calculated from previously reported structures[10,13].

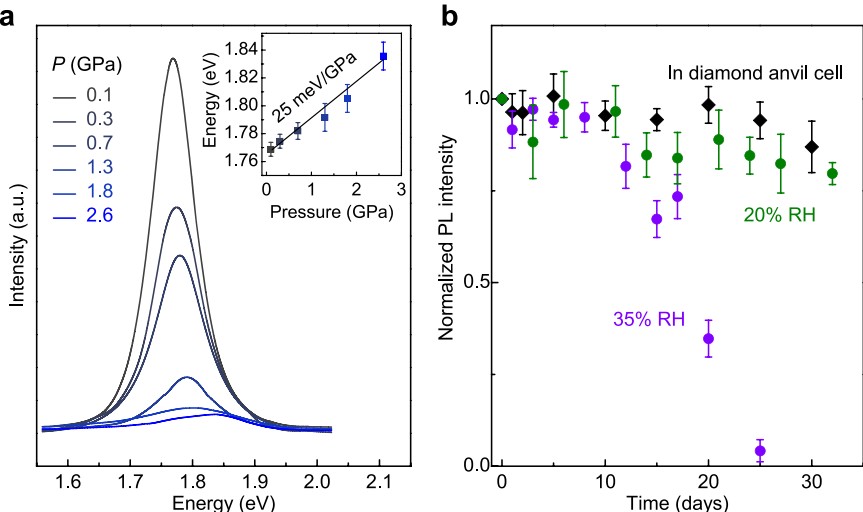

**Fig. 4 PL spectra of re-pressurizing the recovered γ-CsPbI₃ perovskite and PL intensity over time of the preserved γ-CsPbI₃. a** PL spectra collected as a function of pressure at room temperature. Inset: The pressure dependence of the PL energy. A pseudo-Voigt function is used to fit the PL peak that yields the peak position and uncertainty. **b** Normalized PL intensity of the preserved γ-CsPbI₃ perovskite over time in a diamond–anvil cell and in air at 20% and 35% RH, respectively. The error bars are defined as the largest deviation from the average value of the normalized PL intensity collected at different sample positions.

Under compression, the out-of-phase tilt increases rapidly below 0.4 GPa and continues the upward trend at a reduced rate with further compression (0.4–0.8 GPa), followed by a pressure-invariant behavior of staying at ~10° between 0.9 and 1.4 GPa (Fig. 3c). On the other hand, the in-phase tilt increases linearly with pressure up to 1.4 GPa. The similar trend between the out-of-phase tilt and $\Delta E$ below 1.4 GPa, especially between 0.9 and 1.4 GPa, suggests that the out-of-phase tilt contributes the most to the reduction of $\Delta E$ and consequently the preservation of γ-CsPbI₃. At above 1.5 GPa, the in-phase tilt rises sharply, concurrent with the dramatic increase of $\Delta E$. This indicates that at higher pressures, the in-phase tilt could be the main contributing factor for the increase of the total energy of γ-CsPbI₃ that again favors the formation of δ-CsPbI₃.

The compression-directed tilt of $[PbI_6]^{4-}$ octahedra is also reflected from the PL measurements of re-pressurizing the preserved γ-CsPbI₃. The preserved γ-CsPbI₃ exhibits a PL peak at ~1.77 eV at ambient conditions (Fig. 4a), which is comparable with the value of the strain-stabilized γ-CsPbI₃[29] and ~ 60 meV smaller than that of γ-CsPbI₃ synthesized by a nano crystallization method[5]. With re-compressing the recovered γ-CsPbI₃ at room temperature, the PL peak shows a blue shift at a rate of 25 meV/GPa up to 2.6 GPa beyond which the PL signal disappears (Fig. 4a, inset). The pressure response is different from that observed in CsPbI₃ perovskite nanocrystals where the PL at first shifted to lower energies below 0.4 GPa and then to higher energies with further compression[38,39]. These differences could be caused by the different structures and particle sizes of the samples. However, the PL development of a CsPbI₃ perovskite structure as a function of pressure is typically controlled by both the intra-octahedral compression and inter-octahedral tilt, with the former resulting in a red shift and the latter causing a blue shift[39]. The exclusive blue shift observed in the PL of our preserved γ-CsPbI₃ supports the inter-octahedral tilt being significantly modulated at high pressure. The band-structure calculations confirm that the increase of the out-of-phase and in-phase tilt both enlarges the bandgap of γ-CsPbI₃ with the in-phase tilt playing a bigger role (Supplementary Fig. 10).

Further study on the functionality of the preserved γ-CsPbI₃ shows that this perovskite phase has substantial stability to humidity. The PL intensity of the recovered samples remains above 80% of the initial value for more than 30 days and up to 10 days in air at 20% and 35% RH, respectively (Fig. 4b). After degradation in air over time, γ-CsPbI₃ transforms back to the yellow δ-CsPbI₃. With reheating to high temperature at ambient pressure, the preserved γ-CsPbI₃ transforms to the yellow δ-CsPbI₃ phase above 100 °C (Supplementary Fig. 11). With applying pressure at room temperature, Raman measurements indicate that the preserved γ-CsPbI₃ is metastable to at least 5.0 GPa (Supplementary Fig. 11).

In summary, a metastable γ-CsPbI₃ perovskite phase can be synthesized and then preserved back to room temperature via pressure-tuning the tilt of $[PbI_6]^{4-}$ octahedra, given a pressure between 0.1 and 0.6 GPa is applied to the yellow δ-CsPbI₃ followed by heating and rapid quenching. The occurrence of α-CsPbI₃ at high temperatures likely assists the formation of γ-CsPbI₃. Once formed, γ-CsPbI₃ can be retained after releasing pressure to ambient conditions and shows substantial stability in air at 35% RH for up to 10 days. First-principles DFT calculations suggest that pressure directs the out-of-phase and in-phase octahedral tilt that control the relative energy difference between the competing δ- and γ-CsPbI₃ and ultimately stabilizes γ-CsPbI₃. Our study provides key insight into manipulating the phase (meta)stability of halide perovskites through structural control and opens effective pathways to synthesizing metastable phases with improved properties.

## Methods

**Synthesis of δ-CsPbI₃ samples.** The yellow non-perovskite δ-CsPbI₃ samples were synthesized using a solution-based method adapted from literature[42]. Solid PbI₂ (0.46 g, 1.0 mmol) and CsI (0.26 g, 1.0 mmol) were first mixed with aqueous HI (10.0 mL, 7.58 M, stabilized), and then heated to 130 °C with stirring in a closed 20-mL glass vial. The solution was then cooled to room temperature at a rate of 25–35 °C/h by turning off the power of the hot plate. The yellow δ-CsPbI₃ solid powders were collected by vacuum filtration, rinsed with large amounts of diethyl ether, and dried under reduced pressure for 12 h.

**High-pressure and high-temperature apparatus.** High pressure was achieved using a BX-90 type diamond–anvil cell with an anvil culet size of 500 or 600 μm in diameter. A home-made external coiled resistive heater with a resistance of ~2.5 Ω made of KA1 alloy wires (Hyndman Industrial Products) was placed around the T301 stainless-steel gasket and the tip of the diamond anvils to heat up the samples.

The exposed KA1 alloy wires were covered with 940 HT fast cure alumina adhesive for electrical insulation. Asbestos thermal insulation layers were used to cover the remaining area of the diamonds, seats, and the resistive heater to ensure temperature uniformity across the sample.

**Sample loading and reaching high *P–T* conditions**. Powdered δ-CsPbI₃ samples, together with Au powders or a ruby ball as pressure calibrants, were loaded into 250-μm sample chambers in pre-indented T301 stainless-steel gaskets in air. No pressure transmitting medium was used to ensure good contacts between samples and the diamond surface for temperature monitoring. The pressure at the room and the high temperature was calibrated using the equation of state of Au[43,44] or the fluorescence shift from a ruby ball[45]. The temperature was monitored by a K-type thermocouple glued on the diamond pavilion close to the culet. The pressure uncertainty at high temperature was determined based on the temperature fluctuation before and after the XRD and Raman measurements. A higher heating temperature and a longer heating duration favor the conversion to pure γ-CsPbI₃.

**Synchrotron XRD measurements**. High-temperature and high-pressure XRD measurements were performed at beamline 12.2.2 of the advanced light source (ALS), Lawrence Berkeley National Laboratory; and beamline 16-BMD of the Advanced Photon Source (APS), Argonne National Laboratory (ANL). Two-dimensional Debye-Scherrer diffraction rings were collected at a wavelength of $\lambda = 0.4959$ Å (12.2.2, ALS) and $\lambda = 0.4133$ Å (16-BMD, APS) on a Mar345 image plate detector, and integrated using the *Dioptas* software package[46], yielding intensity versus 2θ patterns. The intensity has been normalized for better comparison. The sample-to-detector distance and other parameters of the detector were calibrated using the CeO₂ standard.

**Raman spectroscopy measurements**. Raman spectra were collected using a Horiba LabRam HR Evolution Raman system at the Stanford Nano Shared Facilities (SNSF). A laser excitation wavelength of 633 nm was utilized. A threshold power of ~0.5 mW was kept throughout the measurements to avoid the potential laser-induced heating on the samples. Three accumulations with an exposure time of 10 s per accumulation were done to obtain a good Raman spectrum. Ultra-low-frequency filter kits (ULFK633-17-LR) were used to obtain Raman signals down to 10 cm⁻¹. Before the measurements, the Raman system was calibrated using the Raman mode of a silicon wafer at 520 cm⁻¹.

**PL measurements**. The evolution of PL intensity over time was collected on quenched samples using the Horiba system at SNSF ($\lambda = 532$ nm) and the Renishaw inVia Raman system in Extreme Environments Laboratory at Stanford University ($\lambda = 514.5$ nm). Laser power as low as 0.1 mW and an exposure time of 2 s per measurement were used. The PL peaks were smoothed by using the Savitzking Golay filter to remove the noise and ruby signal. Several positions of the γ-CsPbI₃ samples were monitored with time. The PL intensity of each sample position at a different time was first normalized to that measured at day 1, and then the normalized values were averaged to get a normalized PL intensity versus time curve as shown in Fig. 4b. The PL spectra as a function of pressure were obtained by re-compressing the recovered samples to high pressure at room temperature.

**Humidity tests**. Humidity tests were performed by monitoring the PL intensity of γ-CsPbI₃ over time in air at different levels of RH. The preserved γ-CsPbI₃, along with a hygrometer for humidity calibration were sealed in a glass bottle. The hygrometer was calibrated using the saturated NaOH (~7% RH) and MgCl₂ (~35% RH) solutions at ambient conditions[47]. A 5% RH uncertainty was observed from the hygrometer readings. A RH of 20% was obtained by sealing the sample, the hygrometer, and a small number of desiccants together in a glass bottle.

**First-principles calculations**. DFT calculations for the total energies of the δ- and γ-CsPbI₃ phases under pressure were performed with Quantum Espresso[48]. The input structures were from experimental configurations[10]. The structures at ambient pressure for both phases were calculated under variable cell relaxation. The structure and related total energy at high pressure were then calculated using the relaxed ambient structure as the initial configuration. The pressure was applied by setting the stress to a target value based on the method developed in previous studies[49,50], which has been implemented in Quantum Espresso. PBE exchange-correlation functional was chosen for the exchange and correlation terms[51]. An $8 \times 6 \times 8$ k-grid for the γ phase and a $6 \times 12 \times 3$ k-grid for the δ phase were used. The kinetic energy and charge density cutoff were 75 and 500 Rydberg, respectively. The Gibbs free energy ($G = E + PV - TS$) was calculated, where $V$ is the optimized unit cell volume of δ- and γ-CsPbI₃ at the corresponding pressure and $S$ is the vibrational entropy of each phase. The entropy $S$ was calculated using the following formula:

$$S = -\frac{\partial F}{\partial T} = \frac{1}{2T}\sum_{\mathbf{q},j} h\nu_j(\mathbf{q})\coth\left[\frac{h\nu_j(\mathbf{q})}{2k_BT}\right] - k_B\sum_{\mathbf{q},j}\ln\left(2\sinh\left(\frac{h\nu_j(\mathbf{q})}{2k_BT}\right)\right) \quad (1)$$

where $\nu_j(\mathbf{q})$ is the energy of the $j$th phonon mode at momentum $\mathbf{q}$. Phonon dispersions were calculated at 0 K using Quantum Espresso[48] and Phonopy[52]. Thermal

expansion was neglected in our calculations, and hence the phonon dispersion and phonon density of states at 0 K were used for entropy calculation at high temperatures. The α- and δ-CsPbI₃ phases have been found experimentally to have the same volume expansion coefficient of $\alpha_v = 1.18 \times 10^{-4}$ K⁻¹ (ref. [53]). It is reasonable to assume γ-CsPbI₃ has a similar volume expansion coefficient. Within the temperature range of interest from 200 to 27 °C where γ-CsPbI₃ is metastably preserved during cooling, the volume change is about $\Delta V/V = \alpha_v \times \Delta T = 2\%$ at ambient pressure, and this volume reduction will be smaller at high pressures. Further considering that both the δ- and γ-CsPbI₃ phases experience similar volume changes, we expect our calculations are valid for predicting the relative phase stability. Previous computational work on CsSnI₃ that studied the thermal expansion effect on the Gibbs free energy further validated our assumption[54]. With the current computing resources, their calculations were done by assuming that high temperature only changes the volume but does not change other degrees of freedom, such as tilt angles. The Gibbs free energies of α- and γ-CsSnI₃ showed a similar behavior as a function of temperature with and without considering the thermal expansion effect[54]. The quasi-harmonic approximation can be taken to include the thermal expansion effect. However, it requires constructing a set of test structures with different expanded unit cell volumes and performing phonon calculations for all the test structures to find out the structure with the lowest Gibbs free energy. In the case of γ-CsPbI₃, the structure has many degrees of freedom, including at least three lattice constants, the Pb–I bond length, and the in-phase and out-of-phase octahedral tilt. The [PbI₆]⁴⁻ octahedron also deviates from the ideal geometry in γ-CsPbI₃, adding extra degrees of freedom for structural variations. The large number of degrees of freedom in γ-CsPbI₃ makes it extremely complex to construct the test structures. It is also beyond the current computational capacity to perform phonon calculations that consider the thermal expansion due to the astronomically increased number of test structures.

**Band-structure calculations**. The band-structure calculations were performed using Quantum Espresso with GGA exchange-correlation functional, and a $12 \times 8 \times 12$ k-point grid. The evolution of the bandgap as a function of the in-phase/out-of-phase tilt was calculated by changing one of the tilts and fixing the other to the value at ambient conditions. Through analyzing the high-pressure DFT structures, we found that the in-phase tilt was directly related to the ratio of the two in-plane lattice constants, $a$ and $c$. Therefore, we tuned the in-phase tilt by increasing the lattice constant $a$ by 0.5%, 1.0%, and 2.0% and decreasing $c$ while keeping the unit cell volume and the fractional coordinates of the Pb and I atoms fixed. This way of constructing the structures was found to effectively change the in-phase tilt angle while minimizing the change of the out-of-phase tilt angle by at least an order of magnitude smaller. In the case of tuning the out-of-phase tilt, the apical I atoms were found to always move within the $bc$ plane while changing the out-of-phase tilt. Hence, we tuned the out-of-phase tilt by rotating the octahedron with the internal bond lengths and angles of the [PbI₆]⁴⁻ octahedron being fixed and moving the apical I atoms within the $bc$ plane.

## Data availability

All data that support the findings of this study are present in the paper and the supplementary information. Additional data related to the study are available from the corresponding author upon request.

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

## Acknowledgements

We thank Martin Kunz, Jinyuan Yan, and Andrew Doran of Beamline 12.2.2 at the Advanced Light Source, and Matthew Smith at Stanford University for experimental assistance. This work was supported by the Department of Energy (DOE), Office of Science, Basic Energy Sciences, Materials Sciences and Engineering Division (DE-AC02-76SF00515). Beamline 12.2.2 is a DOE Office of Science User Facility under contract No. DE-AC02-05CH11231. Portions of this work were performed at HPCAT (Sector 16), APS, ANL. HPCAT operations are supported by DOE-NNSA's Office of Experimental Sciences. The APS is a DOE Office of Science User Facility operated for the DOE Office of Science by ANL under Contract No. DE-AC02-06CH11357. Portions of this work were performed at the Stanford Nano Shared Facilities, supported by the National Science Foundation under award ECCS-1542152. First-principles DFT calculations were supported by the resources of the National Energy Research Scientific Computing Center (NERSC), a DOE Office of Science User Facility operated under Contract No. DE-AC02-05CH11231. N.R.W. was partially supported by a Stanford Interdisciplinary Graduate Fellowship.

## Author contributions

F.K. and Y.L. designed the project and wrote the paper. F.K, C.W., J.Y., S.N, W.L.M., and Y.L. conducted the experiments and analyzed the data. N.R.W. synthesized the sample under H.I.K.'s supervision. C.J and T.P.D. performed the calculations. All authors contributed to the discussion and revision of the paper.

## Competing interests

The authors declare no competing interests.
