## [Peer Review File · Nature Communications]

REVIEWER COMMENTS

Reviewer #1 (Remarks to the Author):

In this paper the Authors attempt to use a highP-highT strategy to stabilize CsPbI₃ in the black phase. While the results are of some interest there is nothing particularly exciting to make this paper acceptable for Nature Comm. First of all there is a plenty of work made to stabilize the CsPbI₃ phase (ACS Energy Lett. 2020, 5, 6, 1974–1985) in the perovskite phase with stability time longer than those reported here. Moreover, the stabilization provided by the authors is on bulk materials which has very limited interest for applications while more interesting strategies and results have been obtained for thin films. Moreover, the evidences of a metastable phase (stable for some time at ambient conditions) induced by high pressure on metal halide perovskites are already present in the current literature and, moreover, performed with strategies which allow to scale-up the prepared materials, instead of relying on few crystallites within a DAC. If the Authors could demonstrate their results, for example, into a multianvil apparatus this could be more interesting. Providing the highP-highT phase diagram of CsPbI₃ is not breakthrough and the results reported could be easily anticipated in this phase.

Overall this manuscript does not provide the kind of forefront results required for Nature Comm.

Reviewer #2 (Remarks to the Author):

The manuscript by F. Ke et al. discusses the p-T phase diagram of the all-inorganic halide perovskite CsPbI₃. The present work moves a step ahead compared to other high-pressure studies in halide perovskites by also monitoring the effect of temperature in the crystallization of the metastable black phase (γ -phase) of CsPbI₃. CsPbI₃ is technologically relevant due to its attractive all-inorganic nature, yet it is a problematic compound in the sense that the useful black phase is metastable thereby preventing its application in the fabrication of solar cell devices. In this work, the authors provide for the first time the complete p-T diagram of the compound and what is more, they complement their X-ray structural data with vibrational spectroscopy which is unprecedented for this type of compounds. Optical spectroscopy and DFT calculations complement this very well-written manuscript. What is more, the authors do discover some finite conditions where the metastable phase can “survive” for long periods of time and as a result this work is relevant to applications despite the fundamental nature of the work.

Based on the above, I am happy to recommend acceptance of the manuscript for publication in Nat. Commun. provided the authors address some of the following minor comments and suggestions:

1) In Figure 2 and particularly in panels b and c, some of the reflections seem to have weak intensity or they are missing. As the authors suggest, this is the result of preferred orientation, in accordance to Figure S4. Still it would be very welcome if the authors could discuss the orientation directions, since the lack of diffraction at low angles (particularly those corresponding to (100) and (110) of the α -phase) is highly unusual.

2) In the experimental part of the phonon dispersion calculations, the author mention that they neglected the effects of thermal expansion in their high temperature entropy calculations. This seems to be an approximate method, since it is established that halide perovskites are among the solids with the largest thermal expansion coefficients. Does the thermal expansion effect captured by the change in the unit cell volume alone in the Gibbs free energy equation? It would be nice if the authors could provide some additional discussion or some “control” calculations, showing that this approximation holds for the studied temperature range.

Reviewer #3 (Remarks to the Author):

The paper by Ke & co-workers reports on Preserving a robust CsPbI₃ perovskite phase via pressure-directed octahedral tilt. Currently, stabilizing CsPbI₃ in its well sought-after black phase at room temperature is an active topic as it could eventually allow the possibility to turn to all-inorganic metal-halide systems for better stability. To this end, the authors combined the use of both temperature and pressure to control the phase transition of CsPbI₃ in order to ultimately stabilize it in its metastable γ -CsPbI₃ phase, which is suitable for photovoltaics. From their study, the authors reported a suitable pressure range to ensure the formation of the metastable black γ -CsPbI₃ phase at room temperature and under ambient pressure. The authors used a variety of experimental (XRD, Raman etc..) and theoretical techniques to present and support their findings. Overall, this is a very interesting work as it could bring a valuable synthesis strategy and contribution in the context of halide perovskite based optoelectronics.

Nevertheless, the following questions should be addressed before the paper can be recommended.

- In the XRD results of Fig.2, the diffraction pattern/peaks of the γ -CsPbI₃ black phase at 0.6 GPa (150°C, 30°C) in (b) are quite different from those of the same phase at 0.6GPa (100°C) or 0.5GPa (30°C) in (c). Clearly, these two do not seem to be of the same phase. This needs to be clarified.

- Still regarding the XRD of γ -CsPbI₃: in the 2 θ range between 0 and 20°, there is a relatively intense peak in the XRD plot at about 14° as reported in ref 9 of the manuscript. However, this peak is missing in Fig2b.

On the reverse side, many of the low 2 θ peaks (< 20°) of Fig.2b,c are completely absent in the XRD of ref 9 for γ -CsPbI₃ phase. In the same way, there are peaks below 7.5° in ref. 10 that are not present in Fig2.

The authors need to clarify and lift these ambiguities in the identification of the γ -CsPbI₃ phase and its refinement.

- It is interesting that the authors identified viable routes for applying pressures that stabilize γ -CsPbI₃ phase at room temperature. From Fig. 3, it seems that having the final pressure in the range of 0.1GPa < P < 0.6 GPa is more critical for stabilizing a metastable γ -CsPbI₃ phase than the starting pressure. For instance, did the authors test a scenario where the initial pressure is greater than 0.6 GPa but the final pressure falls within 0.1GPa < P < 0.6 GPa?

- Although, the DFT results seem to point to a stabilizing effect of γ -CsPbI₃ within some pressure range, the total energy of γ -CsPbI₃ still remains higher than that of the delta phase. Any explanation for this?

- The argument that the out-of-plane tilt controls the relative energy differences between delta and γ -CsPbI₃ is not convincing. The out-of-plane angle is steeper in the pressure range of 0 to 0.4GPa but the decrease in the energy does not vary coherently with the same fashion in that pressure range. Similarly, above 1.5GPa, the out-of-plane tilt remains about 10° while ΔE goes up quickly following the in-plane tilt.

To resolve this, the authors should perhaps decouple the two effects by constructing models with imposed in-plane and out-plane tilts. Then, compare the total energy of the respective systems and quantify the contribution of each tilt angle or the tilt that contributes the most.

- To further correlate the PL blue shifts that the authors measured with respect to the tilt angles, the authors should show the variation of the band gaps with respect to the tilt angles from the results of Fig.3c as the pressure varies. This could help ascertain which tilt is more responsible for the observed blue shift. Similar plots (band gap with tilt angles) can be found in the literature.

Minor comments:

- The authors should specify how they applied pressure in the theoretical DFT treatment.
- For reproducibility: can the authors describe more precisely how they computed the entropy term "TS"? the used equations etc.? Also, tabulate the contributions of the calculated entropy terms.
- The out-of-plane angle in ref.10 is 9.9° (it is greater than or equal to 10° in ref. 13) while in Fig.3c those points are less than 8° in the plot. This should be corrected.
- The caption of Supplementary Figure 6: "total energy" should be "total energy difference".
- The authors write "The PL intensity of the recovered samples remains unchanged for more than 30 days and up to 10 days in air at 20% and 35% RH, respectively (Fig. 4b)". What is meant by "remains unchanged"? If it means above 75% of the initial intensity, then it is ok but should be stated in the text.
- When γ -CsPbI₃ degrades over time in air, did the authors check that if it was possible to recover γ -CsPbI₃ phase with their temperature-pressure protocol?
- Finally, when γ -CsPbI₃ degrades in air, does it transform to the delta phase? Or does it decompose to CsI and PbI₂?

In this paper the Authors attempt to use a highP-highT strategy to stabilize CsPbI₃ in the black phase. While the results are of some interest there is nothing particularly exciting to make this paper acceptable for Nature Comm. First of all there is a plenty of work made to stabilize the CsPbI₃ phase (ACS Energy Lett. 2020, 5, 6, 1974–1985) in the perovskite phase with stability time longer than those reported here. Moreover, the stabilization provided by the authors is on bulk materials which has very limited interest for applications while more interesting strategies and results have been obtained for thin films. Moreover, the evidences of a metastable phase (stable for some time at ambient conditions) induced by high pressure on metal halide perovskites are already present in the current literature and, moreover, performed with strategies which allow to scale-up the prepared materials, instead of relying on few crystallites within a DAC. If the Authors could demonstrate their results, for example, into a multianvil apparatus this could be more interesting. Providing the highP-highT phase diagram of CsPbI₃ is not breakthrough and the results reported could be easily anticipated in this phase. Overall this manuscript does not provide the kind of forefront results required for Nature Comm.

Response >> We would like to emphasize that our work does far more than just providing the *P-T* phase diagram of CsPbI₃ (although this is also a valuable contribution). In our original manuscript, the *P-T* phase diagram was shown as a figure in the supplementary information (Supplementary Fig. 2). The novelty and originality of our work are 1) we present the first example of pressure-induced preservation of a desired metastable halide perovskite phase to ambient conditions with excellent stability to moisture and heat; 2) we identify viable pathways and discover a subtle pressure window to access and preserve the metastable γ -CsPbI₃; 3) we reveal the transformation mechanism and present a new strategy for manipulating the phase (meta)stability of CsPbI₃ and halide perovskites through directing the octahedral tilts. We believe that this high-pressure synthesis approach we have developed will open exciting opportunities for accessing metastable phases and creating novel perovskites with improved properties. Below we elaborate on these statements and address the reviewer's comments.

Due to the substantial interest of all-inorganic halide perovskites to the photovoltaic community, there has been work that laid out strategies to stabilize the black CsPbI₃ phases to room temperature that include compositional tuning, nanocrystal growth, solvent and surface treatments, and thermal and strain engineering. However, as discussed in our manuscript as well as in Masi *et al.* (ACS Energy Lett. 5, 1974, 2020), these previous approaches present limitations. We developed a new approach using pressure treatments aided by temperature to preserve the CsPbI₃ perovskite

phase to ambient conditions. The retained bulk CsPbI₃ perovskite phase shows substantially better stability to moisture compared to other bulk γ -CsPbI₃ preserved through solid-state and strain engineering methods (JACS, 141, 11435, 2019 and Science 365, 679, 2019). We recognize that the preserved CsPbI₃ perovskite phase based on nanocrystal growth, and solvent and surface treatments may have a longer stability time. However, the surface defects in nanocrystals act as traps for photogenerated carriers, hindering the charge transport and compromising the optoelectronic properties. Solvent and surface treatments induce impurities that affect the charge transport and optical properties (ACS Energy Lett. 5, 1974, 2020). All of these limitations are detrimental for their application as solar cell materials. Moreover, in contrast to the reported strategies, our method does not require rigorously anhydrous reagents or a moisture-free environment, which offer much flexibility to the sample preparation and synthesis. Furthermore, we revealed the correlation between octahedral tilts and phase (meta)stability and uncovered the underlying mechanism on the preservation of a CsPbI₃ perovskite phase to room temperature, i.e., pressure-directed octahedral tilts that tune the relative energy difference between the γ and δ phase. We believe that this approach of tuning the octahedral tilt can be applicable for preserving the metastable perovskite phases in other systems, such as FAPbI₃ (FA = CH(NH₂)₂⁺) and CsSnI₃, as well.

While our study was conducted on bulk materials, we believe that a similar approach is also applicable on film samples because we prove that mechanistically it is the pressure-directed octahedral tilt that controls the phase metastability. By reducing the dimension to a film or shrinking the grain size to nano particles, the increased surface energy has a positive contribution to the formation of a perovskite structure that may further assist in the preservation of a CsPbI₃ perovskite phase using our developed high-pressure and high-temperature method.

To the best of our knowledge, there has been no report on preserving a desired metastable halide perovskite phase to ambient conditions after pressure treatments. A few previous studies reported pressure-retained bandgap narrowing after decompression in 3D Cs₂AgBiBr₆ (Angew. Chem. 129, 16185, 2017) and 2D hybrid halide perovskites (BA)₂(MA)_{*n*-1}Pb_{*n*}I_{3*n*+1} (BA = CH₃(CH₂)₃NH₃⁺; MA = CH₃NH₃⁺; *n* = 1, 2, 3, 4) which was ascribed to pressure-derived metastable electronic states (ACS Energy Lett. 2, 2518, 2017 and Proc. Natl. Acad. Sci. 115, 8076, 2018). In these cases, well-crystallized halide perovskite phases were used as starting materials. After decompression, the recovered phases had the same crystal structures with identical space groups as the starting phases. The slightly widened inter-octahedral Pb-I-Pb bond angles and varied lattice constants affected the electronic structures and resulted in the narrowed bandgap in the retained materials. A very recent study reported similar changes of bandgap in another series of 2D hybrid halide perovskites A'(MA)_{*n*-1}Pb_{*n*}I_{3*n*+1} [A' = 3-(aminomethyl) piperidinium (3AMP); *n* = 1, 2, 4] after

pressure treatments again due to the slight differences in the lattice parameters. The 2D to 3D structure changes that occurred in the $n = 4$ perovskite after decompression was due to decomposition of $(3\text{AMP})(\text{MA})_3\text{Pb}_4\text{I}_{13}$ into $(\text{MA})\text{PbI}_3$ and other unknown species (Proc. Natl. Acad. Sci. 117, 16121, 2020). Our study presents the first instance of truly preserving a metastable perovskite phase to ambient conditions with substantial materials functionality via a high-pressure and high-temperature route.

Finally, a DAC is an ideal reconnaissance tool for exploring a wide range of P - T conditions. Without the efforts realized in this work, it would be impossible for a multi-anvil apparatus to explore the detailed synthesis routes of the black CsPbI_3 . The P - T pathways we mapped here can be readily passed on to the multi-anvil techniques for large volume synthesis. We have already initiated the collaboration efforts of using a multi-anvil apparatus for scaled-up synthesis and do find that a black CsPbI_3 perovskite phase can be preserved to ambient conditions using the recipe we developed from our DAC study. A systematic study is ongoing.

In summary, we believe that our study is of substantial interest to the readers of *Nature Communications*.

Reviewer #2 (Remarks to the Author):

The manuscript by F. Ke et al. discusses the p - T phase diagram of the all-inorganic halide perovskite CsPbI_3 . The present work moves a step ahead compared to other high-pressure studies in halide perovskites by also monitoring the effect of temperature in the crystallization of the metastable black phase (γ -phase) of CsPbI_3 . CsPbI_3 is technologically relevant due to its attractive all-inorganic nature, yet it is a problematic compound in the sense that the useful black phase is metastable thereby preventing its application in the fabrication of solar cell devices. In this work, the authors provide for the first time the complete p - T diagram of the compound and what is more, they complement their X-ray structural data with vibrational spectroscopy which is unprecedented for this type of compounds. Optical spectroscopy and DFT calculations complement this very well-written manuscript. What is more, the authors do discover some finite conditions where the metastable phase can “survive” for long periods of time and as a result this work is relevant to applications despite the fundamental nature of the work. Based on the above, I am happy to recommend acceptance of the manuscript for publication in Nat. Commun. provided the authors address some of the following minor comments and suggestions:

Response >> We greatly appreciate the reviewer’s acknowledgement of the significance of our work and the recommendation for publication in *Nature*

Communications. Below we have addressed the reviewer's comments in detail.

1) In Figure 2 and particularly in panels b and c, some of the reflections seem to have weak intensity or they are missing. As the authors suggest, this is the result of preferred orientation, in accordance to Figure S4. Still it would be very welcome if the authors could discuss the orientation directions, since the lack of diffraction at low angles (particularly those corresponding to (100) and (110) of the α -phase) is highly unusual.

Response >> We thank the reviewer for this suggestion. After pressure-temperature treatments, the starting fine-grained powders experience significant grain growth and transform into highly textured, coarse grains (Supplementary Fig. 5). The XRD pattern is a result of collective contributions from these grains with different (also preferred) crystallographic orientations. The peak intensities that are mainly determined by the grain size and the orientation of each grain relative to the incident X-ray beam are highly unreliable. Furthermore, at a finite pressure, the introduction of deviatoric stress can also affect the peak intensities that cannot be quantified. All these factors increase the complexity of analyzing the diffraction patterns of the recovered phases, including the preferred orientation.

In fact, the (100) peak at $\sim 4.5^\circ$ and (110) peak at $\sim 6.4^\circ$ of the cubic α -CsPbI₃ phase can be clearly seen in our diffraction patterns (Fig. R1). However, the peak intensities are unreliable due to the reasons discussed above.

Fig. R1. (100) and (110) peaks of α -CsPbI₃ shown in Fig. 2b and 2c as marked with red arrows.

Similarly, for the orthorhombic γ -CsPbI₃ perovskite phase, when plotting the data on the log scale, we do observe several weak peaks at low angles. For example, we can see the (020)/(101) peaks at $\sim 4.6^\circ$ in the revised Supplementary Fig. 3, and the (011) peak at $\sim 4^\circ$ and (200)/(121) peaks at 6.6° in the new Supplementary Fig. 4. In our

experiments, the grain with the [020]/[101] orientation could be very small (Fig. 2b and Supplementary Fig. 3) or there is no [020]/[101] oriented grain (Fig. 2c and Supplementary Fig. 4) in these highly textured, coarse grains relative to the incident X-ray beam, causing the intensities of these peaks to be either weak or missing. This also explains why the intensities of the peaks in the integrated diffraction patterns from different runs (Fig. 2b vs Fig. 2c) look different. We have added this discussion into the revised manuscript and supplementary information.

2) In the experimental part of the phonon dispersion calculations, the author mention that they neglected the effects of thermal expansion in their high temperature entropy calculations. This seems to be an approximate method, since it is established that halide perovskites are among the solids with the largest thermal expansion coefficients. Does the thermal expansion effect captured by the change in the unit cell volume alone in the Gibbs free energy equation? It would be nice if the authors could provide some additional discussion or some “control” calculations, showing that this approximation holds for the studied temperature range.

Response >> We thank the reviewer for the suggestion. For phonon dispersion and Gibbs free energy calculations ($G = E + PV - TS$), a quasi-harmonic approximation can be used to include the thermal expansion effect. At temperature T , a set of test structures with different expanded unit cell volumes will be calculated to find out the structure with the lowest Gibbs free energy. Meanwhile, phonon calculations need to be performed for each of these test structures to account for the entropy term. For materials with simple structures such as diamond and silicon where the only structural variable is the lattice constant, it is within the computational capacity to construct a reasonable number of test structures and perform phonon calculations for all of the test structures. However, in the case of γ -CsPbI₃, the structure has a lower symmetry and many more degrees of freedom. Even if we make reasonable assumptions on the volume expansion at temperature T as we will discuss below, intertwined structural degrees of freedom include at least three lattice constants, the Pb-I bond length, and in-phase and out-of-phase octahedral tilts. Moreover, the [PbI₆]⁴⁻ octahedron deviates from the ideal geometry in γ -CsPbI₃, adding extra degrees of freedom for structural variations. The large number of degrees of freedom in γ -CsPbI₃ makes it extremely complex to construct the test structures. It is also beyond the current computational capacity to perform phonon calculations that consider the thermal expansion due to the astronomically increased number of test structures.

Meanwhile, a previous study on molecular crystals showed that, for Gibbs free energy calculations, the thermal expansion effect on entropy could be partially cancelled by its impact on enthalpy (Acta Cryst. B72, 514, 2016). Another computational work on CsSnI₃ studied the thermal expansion effect on the Gibbs free energy by assuming

that high temperature only changes the volume but does not change other degrees of freedom, such as tilt angles. The Gibbs free energies of α - and γ -CsSnI₃ showed a similar behavior as a function of temperature with and without considering the thermal expansion (Phys. Rev. B 91, 144107, 2015). Therefore, it is generally considered accurate to calculate Gibbs free energies without considering the thermal expansion effect for predicting the phase stability of halide perovskites.

Furthermore, α - and δ -CsPbI₃ have been found experimentally to have the same volume expansion coefficient of $\alpha_v = 1.18 \times 10^{-4} \text{ K}^{-1}$ (J. Phys. Chem. Solids, 69, 2520, 2008). It is reasonable to assume γ -CsPbI₃ has a similar volume expansion coefficient. Within the temperature range of interest from 200 to 27 °C where γ -CsPbI₃ is metastably preserved during cooling, the volume change is about $\Delta V/V = \alpha_v \times \Delta T = 2\%$ at ambient pressure, and this volume reduction will be smaller at high pressures. Further considering that both the δ - and γ -CsPbI₃ phases experience similar volume changes, we expect our original calculations are valid for predicting the relative phase stability.

We therefore used the harmonic approximation to calculate the entropy using the phonon modes of the structure at 0 K. Thermal expansion is neglected in our calculations and the same approximation has also been used in a previous calculation study on γ -CsPbI₃ (ACS Nano 12, 3477, 2018). We have added this discussion in the calculation portion in the methods section.

Reviewer #3 (Remarks to the Author):

The paper by Ke & co-workers reports on Preserving a robust CsPbI₃ perovskite phase via pressure-directed octahedral tilt. Currently, stabilizing CsPbI₃ in its well sought-after black phase at room temperature is an active topic as it could eventually allow the possibility to turn to all-inorganic metal-halide systems for better stability. To this end, the authors combined the use of both temperature and pressure to control the phase transition of CsPbI₃ in order to ultimately stabilize it in its metastable γ -CsPbI₃ phase, which is suitable for photovoltaics. From their study, the authors reported a suitable pressure range to ensure the formation of the metastable black γ -CsPbI₃ phase at room temperature and under ambient pressure. The authors used a variety of experimental (XRD, Raman etc..) and theoretical techniques to present and support their findings. Overall, this is a very interesting work as it could bring a valuable synthesis strategy and contribution in the context of halide perovskite based optoelectronics. Nevertheless, the following questions should be addressed before the paper can be recommended.

Response >> We thank the reviewer for the positive comments on our work. We believe we have addressed all the questions raised by the reviewer and the revised manuscript meets the criteria for its publication in *Nature Communications*.

- In the XRD results of Fig.2, the diffraction pattern/peaks of the γ -CsPbI₃ black phase at 0.6 GPa (150°C, 30°C) in (b) are quite different from those of the same phase at 0.6GPa (100°C) or 0.5GPa (30°C) in (c). Clearly, these two do not seem to be of the same phase. This needs to be clarified.

Response >> The different diffraction patterns of γ -CsPbI₃ in Fig. 2b and 2c are a result of highly textured, coarse grains that form after the heating and cooling cycles. After pressure-temperature treatments, the starting fine-grained powder samples experience significant grain growth and transform to coarse grains with different (also strongly preferred) crystallographic orientations (Supplementary Fig. 5). The peak intensities that are mainly determined by the grain size and the orientation of each grain relative to the incident X-ray beam are therefore highly unreliable. Consequently, the peak intensities in the diffraction patterns of the recovered phase from different runs look different.

While the relative peak intensities are different in Fig. 2b and 2c, all the observed diffraction peaks can be indexed to the γ -CsPbI₃ perovskite phase, not to the α , β or δ phase (see revised Supplementary Fig. 3 and a new Supplementary Fig. 4). Furthermore, both the samples preserved through the routes 2 and 2' (Fig. 2b and 2c) have similar Raman and PL signals that originate from γ -CsPbI₃. All these pieces of evidence support that the preserved samples of Fig. 2b and 2c are the same phase.

We have added more discussion in the revised manuscript to clarify the intensity differences of the diffraction peaks. We have also made changes to Supplementary Fig. 3 and added a new Supplementary Fig. 4 to show the index results of the diffraction patterns in Fig. 2b and 2c to γ -CsPbI₃.

- Still regarding the XRD of γ -CsPbI₃: in the 2θ range between 0 and 20°, there is a relatively intense peak in the XRD plot at about 14° as reported in ref 9 of the manuscript. However, this peak is missing in Fig2b. On the reverse side, many of the low 2θ peaks (< 20°) of Fig.2b,c are completely absent in the XRD of ref 9 for γ -CsPbI₃ phase. In the same way, there are peaks below 7.5° in ref. 10 that are not present in Fig2. The authors need to clarify and lift these ambiguities in the identification of the γ -CsPbI₃ phase and its refinement.

Response >> The diffraction peak at ~ 14° ((002)/(110) peaks of γ -CsPbI₃) in Ref. 9

can be seen at $\sim 4.6^\circ$ in our study after plotting the intensity of Fig. 2b on a log scale (Supplementary Fig. 3). The difference in the two-theta value is due to the different X-ray wavelengths. As discussed above, the nature of the formed γ -CsPbI₃ in our study – coarse grains with strongly preferred orientations – yields unreliable peak intensities, while the samples studied in Refs 9 and 10 were randomly oriented powders. Similarly, the peaks below 20° of Fig. 2b and 2c are also observable in the XRD pattern of Ref. 9 after converting the patterns to the same d -spacing value. Two-theta of 20° in our study ($\lambda = 0.4959 \text{ \AA}$) is about 65.3° in Ref. 9 ($\lambda = 1.5405 \text{ \AA}$). The peak intensities are different due to the reasons discussed above.

The diffraction peaks below 7.5° in Ref. 10 are also present in our diffraction patterns. Two-theta of 7.5° in Ref. 10 is about 9° in our study after converting them to the same d -spacing value (the X-ray wavelength is 0.4139 \AA in Ref. 10 and 0.4959 \AA in our experiments). There are several peaks below 9° in our study. We have added more discussion about the peak intensity differences between our study and previous results in the revised manuscript to lift the ambiguities in the identification of the γ -CsPbI₃ phase.

- It is interesting that the authors identified viable routes for applying pressures that stabilize γ -CsPbI₃ phase at room temperature. From Fig. 3, it seems that having the final pressure in the range of $0.1 \text{ GPa} < P < 0.6 \text{ GPa}$ is more critical for stabilizing a metastable γ -CsPbI₃ phase than the starting pressure. For instance, did the authors test a scenario where the initial pressure is greater than 0.6 GPa but the final pressure falls within $0.1 \text{ GPa} < P < 0.6 \text{ GPa}$?

Response >> While the pressure may shift slightly as temperature is varied during the cycle, for almost all of the runs in our study the final pressure after cooling back to room temperature is nearly identical to the initial pressure before heating. It is technically challenging to have a starting pressure $> 0.6 \text{ GPa}$ but the final pressure falls between $0.1 - 0.6 \text{ GPa}$, because it is not easy to manually control the pressure during a rapid cooling ($\sim 90 \text{ K/min}$) process from high temperatures. Instead, we did the following experimental run, the result of which supports that the final pressure is more critical. We first heat the sample up to $380 \text{ }^\circ\text{C}$ at ambient conditions where CsPbI₃ transforms to the cubic α phase, and increase pressure to 1.1 GPa while keeping the temperature at $\sim 380 \text{ }^\circ\text{C}$. The sample then reverts back to the non-perovskite δ phase. After rapidly cooling down to room temperature, the final pressure is about 0.4 GPa . We find out that γ -CsPbI₃ can be preserved to room temperature and the structural evolution of CsPbI₃ in this run follows Route 2' shown in Fig. 3. We believe the more critical pressure for being able to metastably preserve γ -CsPbI₃ to room temperature is the final pressure applied to the system, i.e. $0.1 - 0.6 \text{ GPa}$.

- Although, the DFT results seem to point to a stabilizing effect of γ -CsPbI₃ within some pressure range, the total energy of γ -CsPbI₃ still remains higher than that of the delta phase. Any explanation for this?

Response >> Since γ -CsPbI₃ is a kinetically trapped metastable phase and does not appear on the P - T phase diagram, it is not unexpected that the energy calculations based on DFT show that γ -CsPbI₃ still has a higher total energy with respect to δ -CsPbI₃. However, the DFT results clearly point to an increased stabilization of the γ phase as pressure falls within the critical window where the total energy difference ($\Delta E = E_\gamma - E_\delta$) reaches its minimum. This reduced ΔE indicates that it is relatively easier to access and preserve γ -CsPbI₃ to room temperature. A similar behavior has also been observed in the strain-stabilized γ -CsPbI₃ film (Science 365, 679, 2019), where ΔE decreases after applying strain that favors the formation of a γ -CsPbI₃ film.

- The argument that the out-of-plane tilt controls the relative energy differences between delta and γ -CsPbI₃ is not convincing. The out-of-plane angle is steeper in the pressure range of 0 to 0.4GPa but the decrease in the energy does not vary coherently with the same fashion in that pressure range. Similarly, above 1.5GPa, the out-of-plane tilt remains about 10° while deltaE goes up quickly following the in-plane tilt. To resolve this, the authors should perhaps decouple the two effects by constructing models with imposed in-plane and out-plane tilts. Then, compare the total energy of the respective systems and quantify the contribution of each tilt angle or the tilt that contributes the most.

Response >> We thank the reviewer for the comment and we have revised the manuscript to reflect that the energy difference varies in a similar fashion with the out-of-phase tilt below 1.4 GPa above which the in-phase tilt contributes more on the sharp rise of the energy difference. As shown in Fig. 3c, the trends between ΔE and out-of-phase tilt below 1.4 GPa are similar, that is, ΔE (out-of-phase tilt) decreases (increases) with pressure below 0.4 GPa, and continues the downward (upward) trend at a reduced rate with further compression (0.4 – 0.8 GPa), followed by a pressure-invariant behavior at 0.9 – 1.4 GPa (Fig. 3c). We can clearly identify the rate-changing loci to be at 0.4 and 0.9 GPa.

We have also attempted to decouple the effect of in-phase and out-of-phase tilts on ΔE . However, during the structural model construction, the unit cell volume (i.e. the lattice constants) and the Pb-I bond length will also be changed while varying only one of the tilts. All of these structural parameters vary dependently and regulate the total energy, making a decoupled study impossible.

- To further correlate the PL blue shifts that the authors measured with respect to the tilt angles, the authors should show the variation of the band gaps with respect to the tilt angles from the results of Fig.3c as the pressure varies. This could help ascertain which tilt is more responsible for the observed blue shift. Similar plots (band gap with tilt angles) can be found in the literature.

Response >> We thank the reviewer for the suggestion. As discussed above, the as-formed γ -CsPbI₃ is composed of highly textured, coarse grains that produce unreliable peak intensities in the diffraction pattern. The structural complexity of γ -CsPbI₃ further excludes us from obtaining precise structural solutions (i.e., atomic coordinates, tilt angles, etc.) with pressure experimentally.

To correlate the PL blue shifts with the tilt angles, we conducted additional calculations on the evolution of the bandgap as a function of the in-phase tilt and the out-of-phase tilt. A previous calculation (Mater. Horiz. 4, 206, 2017) studied the effect of in-phase tilt on the bandgap by assuming the out-of-phase tilt angle to be zero. However, it is far from being accurate in γ -CsPbI₃. According to our DFT calculations, the γ -CsPbI₃ structure is rather complex and all the structural parameters that include the lattice constants, the Pb-I bond length, the in-phase and out-of-phase tilt are correlated. Moreover, the [PbI₆]⁴⁻ octahedron deviates from the ideal geometry in γ -CsPbI₃. All these factors increase the complexity of decoupling the effect of in-phase and out-of-phase tilt on the bandgap. To better tackle the problem and reflect more of a true γ -CsPbI₃ structure, we investigated the effect of in-phase tilt on the bandgap by setting the out-of-phase tilt angle to be the actual experimental value from the previously reported ambient-pressure structure, and vice versa for studying the effect of out-of-phase tilt on the bandgap.

Through analyzing the high-pressure DFT structures, we find that the in-phase tilt is directly related to the ratio of the two in-plane lattice constants a and c . Therefore, we tune the in-phase tilt by increasing the lattice constant a by 0.5%, 1.0% and 2.0% and decreasing c while keeping the unit cell volume and fractional coordinates of Pb and I atoms fixed. This way of constructing the structures is found to effectively change the in-phase tilt angle, while minimizing the change of the out-of-phase tilt angle by at least an order of magnitude smaller. In the case of tuning the out-of-phase tilt, the apical I atoms are found to always move within the bc plane while changing the out-of-phase tilt. Hence, we tune the out-of-phase tilt by rotating the octahedron with the internal bond-lengths and angles of the [PbI₆]⁴⁻ octahedron being fixed and moving the apical I atoms within the bc plane.

The calculations show that the bandgap enlarges with increasing both the in-phase and out-of-phase tilts, and the in-phase tilt has a larger effect on widening the bandgap

(see the new Supplementary Fig. 10). We have added discussion and the calculation details in the revised manuscript.

Minor

comments:

- The authors should specify how they applied pressure in the theoretical DFT treatment.

Response >> The theoretical structure at high pressure was calculated by structural optimization with setting “stress” to a target pressure. The formula to calculate stress in the density functional theory framework was derived by applying infinitesimal homogeneous scaling of the ground state, developed by Nielson and Martin (Phys. Rev. Lett. 50, 697, 1983 and Phys. Rev. B 32, 3780, 1985). In our calculations, this procedure was implemented by performing variable-cell structural optimization and setting “press” to a target pressure in the Quantum Espresso. We have added the related content in Methods.

- For reproducibility: can the authors describe more precisely how they computed the entropy term “TS”? the used equations etc.? Also, tabulate the contributions of the calculated entropy terms.

Response >> The entropy was calculated using the following formula:

$$S = -\frac{\partial F}{\partial T} = \frac{1}{2T} \sum_{q,j} hv_j(q) \coth \left[\frac{hv_j(q)}{2k_B T} \right] - k_B \sum_{q,j} \ln \left(2 \sinh \left(\frac{hv_j(q)}{2k_B T} \right) \right)$$

where $v_j(q)$ is the energy of the j th phonon mode at momentum q .

Details have been added to Methods in the revised manuscript and a new Supplementary Table 1 tabulates the contributions of the calculated entropy terms.

- The out-of-plane angle in ref.10 is 9.9° (it is greater than or equal to 10° in ref. 13) while in Fig.3c those points are less than 8° in the plot. This should be corrected.

Response >> We thank the reviewer for noticing the difference. We have carefully checked the results. The difference is caused by the different definitions of tilt angles. In Ref. 10 and its citation about the definition of tilt angles (ACS Nano 10, 9776, 2016), they used the Pb-Pb-I angle (Fig. R2) to calculate the out-of-plane and in-plane tilts, which reflect the octahedral tilt relative to the axis. In our study, we used a more conventional definition in crystallography – the Glazer notation (Acta. Cryst. B28, 3384, 1972 and Acta. Cryst. B53, 44, 1997) – to quantify the octahedral tilts, in which

the component tilt angles are defined relative to the pseudo-cubic axes of the perovskite structure. We used this definition in our study because it can clearly show the relative tilts among adjacent $[\text{PbI}_6]^{4-}$ octahedra. To clarify our definition, we have changed the original “in-plane tilt” and “out-of-plane tilt” to “in-phase tilt” and “out-of-phase tilt”, respectively. We then calculated the out-of-phase and in-phase tilts of $\gamma\text{-CsPbI}_3$ along the $[101]$ and $[010]$ direction, respectively (Fig. R3). We have also obtained an out-of-phase tilt of 10.16° for our ambient structure if using the definition in Ref 10.

We have added these details of tilt angles in the revised supplementary information (Supplementary Fig. 9).

Fig. R2. The in-plane and out-of-plane tilt angles defined in previous studies (ACS Nano 12, 3477, 2018 and ACS Nano 10, 9776, 2016).

Fig. R3. The out-of-phase and in-phase tilt angles defined in our study. θ represents the torsion angle.

- The caption of Supplementary Figure 6: “total energy” should be “total energy

difference”.

Response >> This has been revised as suggested.

- The authors write “The PL intensity of the recovered samples remains unchanged for more than 30 days and up to 10 days in air at 20% and 35% RH, respectively (Fig. 4b)”. What is meant by “remains unchanged”? If it means above 75% of the initial intensity, then it is ok but should be stated in the text.

Response >> We have revised it to “80% of the initial intensity”.

- When γ -CsPbI₃ degrades over time in air, did the authors check that if it was possible to recover γ -CsPbI₃ phase with their temperature-pressure protocol?

Response >> γ -CsPbI₃ transforms to the δ phase after exposing in air over time and can be recovered with the high-pressure high-temperature approach we developed.

- Finally, when γ -CsPbI₃ degrades in air, does it transform to the delta phase? Or does it decompose to CsI and PbI₂?

Response >> The color of the sample goes back to yellow as γ -CsPbI₃ degrades over time. The Raman and silent PL signals also indicate that γ -CsPbI₃ transforms to the δ phase after exposure in air over time.

REVIEWERS' COMMENTS

Reviewer #1 (Remarks to the Author):

After carefully reading the answers to my criticisms I still remain of my general idea that this paper does not provide the level of novelty in the results requested by Nature Comm. I would like to stress that a paper on the stabilization with pressure of metastable cubic MAPI with induced stable PL effects after pressure application is already out. Moreover, such stabilization, even on a different compound, was demonstrated with a piston cell providing the proper pathway for scale up. The iteration that their way of stabilizing the CsPbI₃ phase can be in some way applied to thin film is hardly to be believed. Overall this paper should be published in a more specialized journal.

Reviewer #3 (Remarks to the Author):

The authors have thoroughly answered all the questions I raised. The paper is now clear and presents an interesting contribution to the synthesis strategy and control of CsPbI₃ based perovskites. Therefore, I recommend its publication in Nature Communications.

As a minor comment, without the need of a new round of review, the authors should specify in their final version of the manuscript that the apparent discrepancies between their work and refs 9, 10 in the 2-teta values of some peaks of gamma-CsPbI₃ is due to the differences in the used X-ray wavelengths of the respective XRD experiments.

December 12th, 2020

REVIEWERS' COMMENTS

Reviewer #1 (Remarks to the Author):

After carefully reading the answers to my criticisms I still remain of my general idea that this paper does not provide the level of novelty in the results requested by Nature Comm. I would like to stress that a paper on the stabilization with pressure of metastable cubic MAPI with induced stable PL effects after pressure application is already out. Moreover, such stabilization, even on a different compound, was demonstrated with a piston cell providing the proper pathway for scale up. The iteration that their way of stabilizing the CsPbI₃ phase can be in some way applied to thin film is hardly to be believed. Overall this paper should be published in a more specialized journal.

Response >> We want to emphasize that we stand firmly on the remarks we laid out in the last response about what is exciting and original in our work. We believe the following paper is what Reviewer #1 referred on “the stabilization with pressure of metastable cubic MAPI with induced stable PL effects after pressure application” (Chem. Commun. 54, 13212, 2018). Again, in this study, the well-crystallized (MA)PbI₃ in a stable perovskite phase was used as the starting material. After decompression, the recovered phases remained in the same crystal structure as the starting phase. “Local structural distortions and atomic configurations” were proposed to explain the slightly reduced unit cell volumes of the pressure-treated samples that contributed to the subtle changes in the optical properties (e.g. bandgap, PL lifetime). Our study presents the first instance of preserving a desired metastable perovskite phase with improved stability to ambient conditions from a non-perovskite phase via a high-pressure and high-temperature route.

Mechanistically, as we have proven, it is the pressure-directed octahedral tilt that controls the phase metastability, rather than the sample dimension. In addition, the surface energy of a sample increases obviously after reducing the dimension to a film or shrinking the grain size to nano particles. The increased surface energy has a positive contribution to the formation of a CsPbI₃ perovskite structure (Science 354, 92, 2016 & Chem. Commun. 53, 232, 2017). We believe our high-pressure approach is applicable on a film sample.

Reviewer #3 (Remarks to the Author):

The authors have thoroughly answered all the questions I raised. The paper is now clear and presents an interesting contribution to the synthesis strategy and control of CsPbI₃ based perovskites. Therefore, I recommend its publication in *Nature Communications*.

As a minor comment, without the need of a new round of review, the authors should specify in their final version of the manuscript that the apparent discrepancies between their work and refs 9, 10 in the 2-teta values of some peaks of gamma-CsPbI₃ is due to the differences in the used X-ray wavelengths of the respective XRD experiments.

Response >> We greatly appreciate the reviewer's acknowledgement of the importance of our work and recommendation for publication in *Nature Communications*. We have added the following statement in the revised manuscript "The apparent differences in the diffraction angle (2θ) values of our diffraction peaks from previous reports^{9,10} are due to the different X-ray wavelengths used in the XRD measurements."